# Zebrafish Models of Induced Lymphangiogenesis: Current Advancements and Therapeutic Discovery

**DOI:** 10.3390/ph18071076

**Published:** 2025-07-21

**Authors:** Srdjan Boskovic, Kazuhide Shaun Okuda

**Affiliations:** 1Department of Biochemistry and Chemistry, School of Agriculture, Biomedicine and Environment, La Trobe University, Bundoora, VIC 3086, Australia; s.boskovic@latrobe.edu.au; 2Centre for Cardiovascular Biology and Disease Research, La Trobe Institute for Molecular Science, La Trobe University, Bundoora, VIC 3086, Australia; 3Organogenesis and Cancer Program, Peter MacCallum Cancer Centre, Melbourne, VIC 3000, Australia; 4Sir Peter MacCallum Department of Oncology, University of Melbourne, Melbourne, VIC 3000, Australia

**Keywords:** zebrafish, lymphatic, lymphatic disease, regeneration, drug discovery

## Abstract

Lymphangiogenesis, the formation of new lymphatic vessels, is essential for embryonic development and the maintenance of tissue fluid balance, as well as for responding to physiological challenges such as injury, inflammation, and oedema. This process is also aberrantly activated in pathological conditions including lymphatic anomalies and cancer. Understanding the molecular and cellular mechanisms regulating induced lymphangiogenesis in various conditions is critical for the development of novel anti- or pro-lymphangiogenic therapeutic strategies. In recent years, the zebrafish has emerged as an important model organism for studying both physiological and pathological lymphangiogenesis. Its optical transparency, conserved lymphatic architecture and signalling pathways, and amenability to genetic manipulation and drug screening make it an especially well-suited model. In this review, we highlight zebrafish models used to investigate induced lymphangiogenesis in the context of regeneration, inflammation, fluid imbalance, and congenital lymphatic anomalies. We will also demonstrate how zebrafish are used to discover new drugs targeting lymphatic vessels under various conditions. Finally, we will discuss the current limitations of using zebrafish to model induced lymphangiogenesis and highlight potential future directions. The findings presented in this review underscore the undeniable value the zebrafish model brings to lymphatic research and therapeutic discovery.

## 1. Introduction

The vertebrate lymphatic system comprises a hierarchical network of specialised lymphatic vessels present in most tissues and plays important roles in fluid homeostasis, immune surveillance, and dietary lipid absorption [1]. Lymphatic vessels vary in size and structure, starting with small-diameter, blind-ended initial lymphatic vessels that absorb interstitial fluid. This lymph is then transported through pre-collecting and collecting lymphatic vessels, passes through lymph nodes, and ultimately drains into the blood circulation. Unidirectional flow through the lymphatic system is maintained by lymphatic valves, which prevent the backflow of the lymph [2,3]. Lymph fluid within lymphatic vessels contains various macromolecules drained from tissues, as well as components of the immune system. This allows an opportunity for immune cells to come in contact with immunogens drained from organs and peripheral tissues and mount an immune response [4]. Lymphatic development is predominantly driven by the vascular endothelial growth factor C (VEGF-C)/VEGF receptor 3 (VEGFR3) signalling and downstream Mitogen-activated protein kinase (MAPK)—Extracellular Signal-Regulated Kinase (ERK) and Phosphoinositide 3-kinase (PI3K)—Protein kinase B (AKT) pathways. This signalling cascade promotes lymphatic sprouting and migration, and also induces the expression of prospero homeobox protein 1 (PROX1), a master regulator of lymphatic differentiation [5,6,7,8,9,10,11,12,13].

Pathologies of the lymphatic system can be broadly categorised into two types: insufficient or excessive lymphangiogenesis. A lack of lymphatic vessels can be caused by inactivating mutations in genetic pathways essential for lymphatic development (e.g., VEGF-C and VEGFR3) or damage to lymphatic vessels, resulting in lymphedema [14,15,16]. This condition is associated with tissue fluid accumulation due to insufficient fluid drainage and is accompanied by complications such as chronic inflammation and recurrent secondary infections [17,18]. On the other hand, gain-of-function mutations that result in constitutively active lymphangiogenic signalling lead to lymphatic vessel hyperplasia, ectopic lymphatic vessel formation, and lymphatic lesions [19,20]. These phenomena are known as lymphatic malformations (LM) or complex lymphatic anomalies (CLA), depending on whether they are localised (LM) or multifocal (CLA). They represent a heterogenous group of diseases with overlapping symptoms, classified according to clinical manifestations. These include small or large fluid-filled cysts, vascular lesions infiltrating tissues including the bone, defects in central collecting channels, as well as leakage of lymph into thoracic or abdominal cavities or externally [21,22,23]. Known causative mutations of LM and CLA are activating mutations in components downstream of VEGF-C/VEGFR3 signalling that are important for lymphatic endothelial cell (LEC) proliferation, survival, and vessel growth. Such mutations have been identified in *PIK3CA* [24,25,26], neuroblastoma RAS viral oncogene (*NRAS*) [27,28], Kirsten rat sarcoma virus (*KRAS*) [29,30], and A-Raf proto-oncogene (*ARAF*) [31]. Beyond lymphatic diseases, excessive lymphangiogenesis can also contribute to pathology in other settings. One example is corneal neovascularisation, where blood and lymphatic vessels invade the normally avascular cornea. This leads to increased inflammation, exacerbating the pathology of various ocular diseases such as glaucoma and uveitis [32]. Excessive lymphangiogenesis can also lead to organ transplant rejection, as new vessels draining the transplant attract immune cells through chemokine signalling and cause host lymphocytes to mount a specific alloantigenic response [33,34]. Cancer-associated lymphangiogenesis is an important aspect of many malignancies, as cancer cells actively recruit lymphatic vessels and use them to metastasise [35,36]. However, increased lymphangiogenesis can have beneficial effects as well. During pulmonary and skin infection in mice, the increased formation of lymphatic vessels is induced by VEGF-C secreted by epithelial cells, dendritic cells, macrophages, and neutrophils [37,38]. The increase in lymphatic coverage promotes drainage of accumulated fluid and immune cell trafficking to the infected tissue, preventing oedema and aiding in infection/inflammation resolution. Likewise, tissue injury is a strong stimulus for lymphangiogenesis, as can be seen after ischemic injury of the heart [39,40], eye injury [41], and skin wounds [42]. In these scenarios, lymphangiogenesis is beneficial and is even considered as a strategy to promote regeneration in chronic wounds [43]. Particularly interesting is the role of lymphatics in chronic and acute inflammation across various contexts. In these conditions, lymphatics have a complex relationship with local tissues and pro-lymphangiogenic immune cells, and can be either beneficial or contribute to further pathologies (reviewed in [44,45,46]). Overall, the context dependent roles of induced lymphangiogenesis warrant better understanding to guide future development of anti- or pro-lymphangiogenic therapeutic strategies for various pathologies.

## 2. Zebrafish as a Model for Lymphatic Research

Induced lymphangiogenesis has been studied using various rodent models of skin wounding and inflammation [42], respiratory and skin bacterial infection [37,38], cancer [42,47,48], and lymphatic diseases [26,49,50,51,52]. Another popular animal model for studying lymphatics is the zebrafish. This small teleost fish, which entered research laboratories in the early 1970s, has made considerable contributions to developmental biology [53,54,55,56]. Physiological and genetic similarities to mammals justify the use of zebrafish as a vertebrate model for human disease conditions [57,58]. Numerous zebrafish models have been developed to study cardiovascular diseases [59,60,61,62,63], cancer [64,65,66,67], neurological disorders [68,69], muscle diseases [70,71,72], infectious diseases [73,74,75,76], and other conditions [77]. Zebrafish have also been increasingly used for drug discovery [78,79,80]. Importantly, zebrafish show a great level of similarity in lymphatic development to mammals. They possess functional lymphatic vessels capable of draining fluid from tissues [6,13,81], and a subset of these vessels contains lymphatic valves that ensure the unidirectional flow of lymph [82,83,84]. Although zebrafish are currently considered to lack lymph nodes, immune cells can travel through their lymphatic vessels [85,86,87,88], and lymphoid organs covered by a lymphatic vessel plexus (termed axillary lymphoid organs) exhibiting features that resemble mammalian lymph nodes have recently been discovered [89]. This indicates that zebrafish lymphatics may also play essential roles in immune surveillance. Importantly, molecular mechanisms driving mammalian lymphangiogenesis are well conserved in zebrafish [7,9,11,90,91]. Studies using zebrafish have identified many evolutionarily conserved genes driving lymphatic development [92,93,94,95,96,97,98,99,100,101,102,103]. A notable example with clinical translation is the identification of collagen and calcium-binding EGF domains 1 protein (CCBE1) as essential for lymphatic development and mutated in human primary lymphoedema Hennekam syndrome [96,97]. Zebrafish are also used to model other primary lymphoedema subtypes including Milroy disease (*fms-related receptor tyrosine kinase 4* (*flt4*, zebrafish orthologue of *VEGFR3*) mutant) [104], Milroy-like disease (*vegfc* mutant) [105,106], and Emberger syndrome (*GATA-binding protein 2a* (*gata2a*) mutant) [82,83]. Coupled with their amenability to various imaging modalities [86,107,108], genetic manipulation [83,109,110], and an array of established transgenic reporter lines with labeled blood and lymphatic vessels (Figure 1A) [81,85,97,111,112,113], zebrafish offer a powerful platform for elucidating mechanisms associated with lymphatic development and diseases. Lymphatic development in zebrafish has been extensively reviewed in the literature [35,56,90]. Briefly, zebrafish lymphatic formation initiates at around 32 h post-fertilisation when Prox1a-positive endothelial cells begin to emerge in veins (Figure 1B) [7,11,103]. These Prox1a-positive endothelial cells leave the veins by 36 h post-fertilisation and give rise to parachordal LECs at the horizontal myoseptum in the trunk (Figure 1C,D) and LECs in the facial lymphatic sprout in the craniofacial region. These LECs continue to proliferate and migrate, forming mature trunk (Figure 1E,F) and facial lymphatic vessels by 5 days post-fertilisation [99,107,114,115,116,117]. A subset of facial lymphatic LECs also originate from non-venous angioblasts and contribute to the development of both lymphatic (lateral facial lymphatic) and arterial (hypobranchial artery) vessels [81,118]. Zebrafish also have an extensive intestinal lymphatic network, first visible at approximately 4 days post-fertilisation, that covers the intestine by 15 days post-fertilisation. The exact origins of the intestinal lymphatics in zebrafish remain to be fully elucidated. As zebrafish transition through the larval, juvenile, and adult stages, increasingly specialised lymphatic vessels emerge, such as spinal, superficial (Figure 1G), cardiac, and meningeal lymphatics [85,86,88,119,120,121,122]. While most of these late-forming lymphatic vessels originate from pre-existing lymphatics, further studies are needed to determine whether LECs of non-venous origin also contribute to their development.

Recently, zebrafish have gained prominence acting as models for studying diseases and conditions associated with induced lymphangiogenesis (see Table 1 and Figure 1A) [30,31,86,87,108,119,121,123,124,125,126,127,128,129]. In this review, we will highlight these studies and discuss how these models are being used to identify potentially clinically relevant therapeutic leads and targets.

## 3. Cancer, Inflammation, Meningeal, and Oedema-Associated Lymphangiogenesis

The first study to use zebrafish as a model for induced lymphangiogenesis was by Hoffman and colleagues, who showed that the xenotransplantation of mouse melanoma B6 cells stimulates lymphatic capillary development in *vegfc*-depleted fish (Table 1) [123]. While the authors show evidence that these vessels are not blood vessels, questions remain about the identity of lymphatic capillaries characterised in this study as a lymphatic-specific marker was not used to confirm their identity. The first model that visualised induced lymphangiogenesis using a zebrafish lymphatic reporter transgenic was conducted by Okuda and colleagues, who showed that dextran sodium sulphate (DSS) or 2,4,6-trinitrobenzenesulfonic acid (TNBS)-induced inflammation is accompanied by a quantifiable increase in both number and length of intestinal lymphatic vessels (Table 1) [124]. This increase is Vegfr signalling-dependent and is accompanied by the recruitment of *vegfc*-expressing macrophages to the intestine. Selective macrophage ablation attenuated inflammatory lymphangiogenesis in zebrafish, showing that zebrafish macrophages can stimulate lymphangiogenesis like in mammals. Interestingly, angiogenesis is not induced in this model, showing a difference to mammalian inflammatory models [130,131]. Nevertheless, the model clearly shows that lymphangiogenesis is induced in inflammatory settings in zebrafish and could be used to understand mechanisms that drive this process.

The breakthrough discovery of a functional meningeal lymphatic network in zebrafish, which expresses common lymphatic markers, opened new opportunities for the zebrafish model [86]. Zebrafish intracranial lymphatic vessels line the inside surface of the skull cap, drain the subdural and subarachnoid space, and traffic immune cells. Furthermore, meningeal lymphangiogenesis in zebrafish can be stimulated by the intracranial administration of recombinant human VEGF-C protein (Table 1). As multiple studies showed the benefits of enhancing meningeal lymphangiogenesis in treating human diseases such as brain cancer and neurodegenerative disorders, the zebrafish model could prove valuable in understanding how therapeutic meningeal lymphangiogenesis can be induced [132,133,134].

Zebrafish have mechanisms for regulating osmotic balance which allows them to adapt to changing water environment [135]. However, in cases of extreme osmotic stress, such as sudden changes in water osmolarity during development, balance is not established quickly enough, leading to the development of oedema in embryos [125]. Taking advantage of this, Olayinka and colleagues recently showed that oedema induces Vegfr-dependent lymphangiogenesis in zebrafish, which is required for the resolution of oedema (Table 1). To induce oedema, they temporarily incubated zebrafish embryos in hypertonic solution, resulting in measurable oedema formation which eventually subsides. Increased lymphangiogenesis was observed in the larvae that recovered, with increased LEC number, enlarged lymphatic vessels, and the occurrence of ectopic lymphatic sprouts. Lymphatic vessel expansion was preceded by an overall transcriptional change in endothelial cells, with the upregulation of lymphangiogenic markers such as *flt4* and lymphatic vessel endothelial hyaluronic receptor 1b (*lyve1b*). Selectively blocking Vegfr signalling, and thereby lymphangiogenesis, using known inhibitors such as SAR131675 and SKLB1002 impaired oedema recovery rate. The lymphangiogenic response to oedema in this model is similar to VEGFR signalling-induced lymphangiogenesis in response to interstitial fluid accumulation seen in mice [136], highlighting the conserved role of zebrafish lymphatics in maintaining fluid homeostasis. Of note, the model highlights the role of lymphatics for managing fluid balance even at this early stage of development (3 to 8 days post-fertilisation) and presents an accessible way to study oedema-induced lymphangiogenic response.

## 4. Tissue Regeneration-Associated Lymphangiogenesis

Unlike in mammals, the zebrafish heart has a remarkable regenerative capacity, being able to replenish lost ventricular cardiomyocytes and resolving scars within 90 days [137]. Multiple injury models have been described so far, such as ventricle apex amputation [62], ventricle cryo-injury [138], and genetic ablation using cardiomyocyte-targeted diphtheria toxin A chain [139], each model with its own characteristics. As angiogenesis is a critical component of the regenerative response following ventricle cryoinjury [140,141,142], several studies investigated the roles of lymphatics in cardiac regeneration [119,121,126,127]. Ventricle injury is followed by an increased sprouting and expansion of lymphatic vessels [119,121,126,127] on the ventricle surface (Table 1). Interestingly, the extent of growth depends on injury, being more pronounced after cryoinjury than amputation [119,121]. Compared to amputation injury, cryoinjury leads to vastly greater necrosis, apoptosis, and inflammation in the tissue. This results in a higher and prolonged *vegfc* expression in thrombocytes and endocardial and myocardial cells in the immediate vicinity of the wound [127]. Overexpressing *vegfc* in the zebrafish heart just prior to injury improved scar resolution. In accordance with the protective role lymphatics play after heart injury, a lack of lymphatic vessels (*vegfc*/*vegfd* double mutants or inhibition of Vegfc signalling using soluble Flt4) led to increased scaring and impaired regeneration despite the fact that lymphatics do not affect cardiomyocyte proliferation in zebrafish [119,121,126]. In the context of injury, lymphatics were proposed to provide a pro-regenerative environment by facilitating the clearance of cellular debris and inflammation. Similarly, ischemic injury such as myocardial infarction leads to a prolonged lymphangiogenic response in mice and rats, with an increase in the number of small diameter vessels and upregulation of *Vegfr3*, *Lyve1b*, and *Prox1* [40,143]. Importantly, stimulating lymphangiogenesis in mice by introducing recombinant VEGF-C in ischemic hearts results in an increased number of lymphatic vessels [40,143], improved inflammation resolution [144], and heart function. The evolutionarily conserved roles of lymphatics in stimulating cardiac regeneration clearly points to the potential of using therapeutic lymphangiogenic approaches for cardiac diseases such as cardiac ischemia.

In a landmark study by Das and colleagues, zebrafish lymphatic vessels were shown to be able to transdifferentiate into secondary vessels (specialised blood vessels found in ray-finned fish [145,146]) with blood vessel function (blood circulation) and blood endothelial cell marker expression (*fms-related receptor tyrosine kinase 1* (*flt1*), *kinase insert domain receptor like* (*kdrl*) and *SRY-box transcription factor 17* (*sox17*)) [108]. They showed that fin resection induces lymphangiogenesis towards the regenerating fin, followed by the transdifferentiation of these lymphatic vessels to secondary vessels (Table 1). Importantly, secondary vessels derived from lymphatic vessels are required for proper fin bone formation, as *flt4* mutants lacking lymphatic vessels exhibit prominent fin bone malformations. This occurs despite their ability to form blood vessels covering the same area from pre-existing blood vessels. While this phenomenon may be exclusive to ray-finned fish (such as zebrafish), it is worth noting that mammalian LECs can also acquire blood endothelial phenotype with haemogenic capacity when *Prox1* expression is reduced [147,148] or when blood flow is driven through lymphatic vessels [149]. The evolutionarily conserved plasticity of LECs should be kept in mind for future therapeutic lymphangiogenesis strategies; the zebrafish fin regeneration model offers a highly accessible platform for gaining insights into this phenomenon.

Live imaging in zebrafish embryos and larvae is relatively easy to perform [150], and recent progress has enabled continuous imaging of adult fish under anaesthesia for up to 20 h [87]. A setup developed by Castranova and colleagues also allowed multiple 3.5 h-long live imaging sessions of superficial wounds in adult zebrafish. In these experiments they observed neutrophil trafficking through lymphatics and accumulation at the wound area starting within the first 3 h after injury. These became prominent by 2 days post-injury (dpi), followed by neutrophil clearance and lymphatic network overgrowth by 7 dpi (Table 1). This study clearly shows the potential of using adult zebrafish to image induced lymphangiogenesis in various diseases and conditions.

## 5. Lymphatic Disease Modelling in Zebrafish

Several studies have used the zebrafish to model lymphatic anomalies. Li and colleagues identified a causative c.2334+1G>C mutation in *EPHB4* in a patient with a central conductive lymphatic anomaly (CCLA) [128]. The mutation activates a cryptic splice site, leading to a frame shift and a loss-of-function mutation in the EPHB4 receptor. Targeting a corresponding splice site in zebrafish by morpholino injection caused the fusion of vessels in the caudal vascular plexus and excessive lateral branching of intersegmental lymphatic and blood vessels (Table 1). Mechanistically, the morpholino targeting *EPHB4* in zebrafish, as well as loss-of-function allele knock-in in HEK293T cells, caused an elevated phosphorylation of mTORC and p70S6 kinase. PI3K/mTORC1 inhibitors rapamycin and BEZ235 and, surprisingly, mitogen-activated protein kinase kinase (MEK) inhibitors U0126 and cobimetinib, rescued the *EPHB4* morpholino-induced vascular phenotypes in zebrafish (Table 2).

KRAS, a member of the RAS family, is able to bind to guanosine triphosphate (GTP) and activate the MAPK signalling cascade to regulate cell growth and proliferation [151]. Sheppard and colleagues focused on two pathological variants of KRAS found in patients with CCLA, p.G12D, and p.G13D [30]. Zebrafish larvae with transient, mosaic lymphatic expression of pathological KRAS variants exhibited fusion of the thoracic duct (TD) and the cardinal vein, tumour-like lymphatic expansion, and pericardial oedema by 5 days post-fertilisation (Table 1). This larval CCL model was then used for screening a set of MEK inhibitors for their ability to prevent the observed phenotypes and identified cobimetinib, AZD8330, pimasertib, and TAK-733 as potential therapeutics (Table 2). Expressing the activating KRAS variants in human dermal lymphatic endothelial cells (HDLECs) led to loss of actin cytoskeletal organisation, increased cellular extensions and protrusions, induced sprouting, and increased phospho-ERK levels. MEK inhibitors trametinib and binimetinib were effective in reverting these cellular phenotypes in vitro.

Activating p.Q61R mutation in NRAS, another member of RAS family upstream of PI3K/AKT/MYC and MEK pathways, was found in a patient diagnosed with generalised lymphatic anomaly and multiple patients diagnosed with Kaposiform lymphangiomatosis (KLA), an aggressive subtype of GLA [27,28,152]. LECs isolated from a patient with GLA showed increased survival and proliferation, an inability to properly form vascular networks, and elevated AKT and ERK phosphorylation levels which could be lowered with rapamycin and trametinib, respectively [28]. In a recent preprint, Bassi and colleagues targeted the expression of NRAS p.Q61R to zebrafish lymphatics and veins at 24 h post-fertilisation using a tamoxifen-inducible *Tg*(*lyve1b:Gal4^ERT2^*);*Tg*(*UAS:hNRASp.Q16R*) line and observed altered LEC morphology, dilated TD, increased body volume, and the formation of pericardial oedema in 5 days post-fertilisation larvae (Table 1) [129]. Trametinib treatment blocked the pathological effects of NRAS p.Q61R, with no signs of swelling or oedema and the reduced dilation of TD in treated larvae (Table 2). Using body size as a readout, authors performed a drug screen based on high-content imaging coupled with artificial intelligence (AI) to identify other potential therapeutics countering the effects of NRAS p.Q61R in zebrafish. This screen pointed to GSK690693 (pan-AKT inhibitor), Verapamil (calcium channel blocker), and cabozantinib (receptor tyrosine kinase inhibitor targeting MET, VEGFR2, FLT3 and c-KIT) as potential hits, with cabozantinib and GSK690693 also effective in a spheroid-based sprouting model using NRAS p.Q61R-harbouring cells isolated from a KLA patient (Table 2).

An impactful study that demonstrated the translational potential of the zebrafish model was reported by Li and colleagues, who identified an activating p.S214P mutation in *ARAF* in two unrelated patients diagnosed with CCLA [31]. This mutation in a highly conserved phosphorylation site results in a constitutively active ARAF at the cell membrane and activation of ERK1/2. Overexpressing human p.S214P ARAF in zebrafish lymphatics using manose receptor C type 1 (*mrc1a*) promoter led to the dilation of lymphatic vessels, including the TD, accompanied by an increase in phospho-Erk level (Table 1). Of note, TD dilation was also seen in one of the patients. Cobimetinib, a MEK inhibitor, rescued the phenotype seen in zebrafish (Table 2). In accordance with this, cultured human dermal lymphatic cells (HDLEC) expressing ARAF p.S214P showed an increase in lymphangiogenic sprouting in a spheroid-based assay in the absence of VEGF-C, reduced VE-cadherin accumulation at the membrane, and alterations in actin organisation. Treatment with trametinib, the same class of MEK inhibitor as cobimetinib, reverted this phenotype, indicating its potential as a therapeutic. As both models show the therapeutic effectiveness of MEK inhibitors in reverting the mutated ARAF phenotypes, an off-label treatment was approved in a young CCLA patients harbouring the same activating *ARAF* mutation. Trametinib administration for just one year resulted in a striking recovery, with improvements in pulmonary function, fluid retention reduction, and lymphatic network remodelling (Table 2). These studies clearly demonstrate the potential of using the zebrafish model for accelerated in vivo discovery of therapeutic avenues for patients with lymphatic anomalies.

**Table 2 pharmaceuticals-18-01076-t002:** Drugs shown to affect lymphangiogenesis in zebrafish models.

Drug	Mechanism of Action	Zebrafish Phenotype	Model	Mammalian Translation	Reference
Cobimetinib	MEK inhibitor	Rescue of *ephb4* morpholino-induced vascular phenotypes	*ephb4* morpholino-induced CCLA	Not tested	[128]
		Reduction in oedema	KRAS p.G12D/p.G13D-induced CCLA	Note tested	[30]
		Rescue of TD morphology	ARAF p.S214P-induced CCLA	Not tested	[31]
BEZ235	PI3K and mTOR inhibitor	Rescue of *epnh4* morpholino-induced vascular phenotypes	*ephb4* morpholino induced CCLA	Not tested	[128]
Binimetinib	MEK inhibitor	Reduction in oedema	KRAS p.G13D induced CCLA model	Rescues the phenotypes in HDLECs expressing KRAS p.G12D/p.G13D and reduces sprout length in organoid model	[30]
AZD8330	MEK inhibitor	Reduction in oedema	KRAS p.G12D induced CCLA model	Not tested	[30]
Pimasertib	MEK inhibitor	Reduction in oedema	KRAS p.G12D induced CCLA model	Not tested	[30]
TAK-733	MEK inhibitor	Reduction in oedema	KRAS p.G12D induced CCLA model	Not tested	[30]
Trametinib	MEK inhibitor	Reduction in body swelling, pericardial oedema, and TD dilation	NRAS p.Q61R induced GLA/KLA model	Reduces sprouting of isolated NRAS p.Q61R patient LECs in spheroid-based assay Rescues the phenotypes in ARAF p.S214P expressing HDLECs and was used to treat an ARAF p.S214P-induced CCLA patient. Rescues the phenotypes in HDLECs expressing KRAS p.G12D/p.G13D and reduces sprout length in organoid model	[30,31,129]
GSK690693	AKT inhibitor	Reduces embryo swelling and TD dilation	NRAS p.Q61R induced GLA/KLA model	Reduces sprouting of isolated NRAS p.Q61R patient LECs in spheroid-based assay	[129]
Verapamil	Calcium channel blocker	Reduces embryo swelling and TD dilation	NRAS p.Q61R induced GLA/KLA model	No effect on isolated NRAS p.Q61R patient LECs in spheroid-based assay	[129]
Cabozantinib	Receptor tyrosine kinase inhibitor (MET, VEGFR2, FLT3, c-KIT)	Reduces embryo swelling and TD dilation	NRAS p.Q61R induced GLA/KLA model	Reduces sprouting of isolated NRAS p.Q61R patient LECs in spheroid-based assay	[129]
Kaempferol	VEGFR2/VEGFR3 kinase activity inhibition	Inhibits lymphatic sprouting, resulting in impaired TD formation	Developmental lymphangiogenesis	Inhibits VEGF-C-induced lymphatic growth in a mouse Matrigel plug assay, and reduces tumour-induced lymphangiogenesis and lymph node metastasis in mice	[153]
Leflunomide	Dihydroorotate dehydrogenase inhibitor	Inhibits lymphatic sprout migration and morphology, resulting in impaired TD formation	Developmental lymphangiogenesis	Inhibits VEGF-C-induced lymphatic growth in a mouse Matrigel plug assay	[153]
A77 1726 (Leflunomide active metabolite)	Dihydroorotate dehydrogenase inhibitor	Reduces number of secondary sprouts, resulting in impaired TD formation	Developmental lymphangiogenesis	Not tested	[153]
Cinnarizine	Calcium channel blocker	Impairs TD formation by inhibition of lymphangiogenesis after secondary sprouting	Developmental lymphangiogenesis	Not tested	[153]
Flunarizine	Calcium channel blocker	Apoptosis of parachordal lymphatic endothelial cells	Developmental lymphangiogenesis	Inhibits VEGF-C-induced lymphatic growth in a mouse Matrigel plug assay. Ineffective at suppressing tumour-induced lymphangiogenesis and lymph node metastasis in mice	[153]
CDF	Inhibition of VEGF-C-induced ERK phosphorylation	Inhibits lymphatic sprouting and migration, leading to impaired lymphatic development. Blocks endothelial cell proliferation in a *vegfc*-induced model	Developmental lymphangiogenesis	Reduces VEGF-C-induced phospho-ERK levels in HMVEC-dLy-Ad	[154]
Toluquinol	Inhibition of VEGF-C-induced phosphorylation of VEGFR3, AKT and ERK	Inhibits TD formation	Developmental lymphangiogenesis	Reduces viability, migration, tube formation, and sprouting of HMVEC-dLy-Ad. Reduces lymphangiogenesis in mouse explanted lymphatic ring, ear sponge, and corneal neovascularization assays	[155]

## 6. Anti-Lymphangiogenic Drug Discovery Using Zebrafish

As discussed above, zebrafish lymphatic disease models have been used to identify various anti-lymphangiogenic drugs with potential clinical relevance (see Table 2) [30,31,129]. An alternative strategy for anti-lymphangiogenic drug discovery in zebrafish is to screen for compounds that inhibit developmental lymphangiogenesis. Due to having a well-defined lymphatic morphology and development which is accessible to imaging and quantification (Figure 1B–G), developmental lymphangiogenesis offers the most straightforward approach for screening anti-lymphangiogenic drugs in zebrafish [79,80]. Using this approach, Astin and colleagues screened the Prestwick Chemical Library for compounds inhibiting lymphangiogenesis [153]. The screen relied on a combination of *lyve1b* mRNA in situ hybridisation staining and imaging of zebrafish lymphatic transgenic reporter lines to identify drugs specifically inhibiting developmental lymphangiogenesis. This led to the identification of kaempferol, leflunomide, cinnarizine, and flunarizine as promising candidates (Table 2). Although all four drugs inhibited TD formation, timelapse imaging of treated embryos showed this likely occurs through different mechanisms. Kaempferol and leflunomide affected lymphatic sprouting either by completely preventing it or by causing abnormal spout morphology and migration. Consistent with this observation, kaempferol was shown to inhibit VEGFR kinase activity, which is essential for lymphatic sprouting. Cinnarizine and flunarizine affected lymphangiogenesis at later stages, with flunarizine shown to cause the apoptosis of parachordal LECs. Kaempferol, leflunomide, and flunarizine also inhibited VEGF-C-induced lymphatic vessel growth in a mouse Matrigel plug assay. In a mouse orthotopic breast cancer xenograft model, kaempferol was shown to inhibit tumour-associated lymphangiogenesis and lymph node metastasis but not tumour growth and metastasis to the pancreas and diaphragm. In contrast, flunarizine failed to inhibit tumour-associated lymphangiogenesis and instead promoted tumour growth and lymph node metastasis. These results show that not all anti-lymphangiogenic drugs can be used to limit tumour lymphangiogenesis and metastasis, while also highlighting the need to combine anti-tumour lymphangiogenic agents with anti-cancer or anti-angiogenic therapies to effectively suppress tumour metastasis. Another example of anti-lymphangiogenic drug discovery in zebrafish is the identification of curcumin analogue 3,4-difluorobenzocurcumin (CDF) [154] (Table 2). CDF treatment at 24 h post-fertilisation, before lymphatic precursors emerge from the veins (Figure 1B), reduced the number of LECs in both the facial and trunk lymphatic networks at 5 days post-fertilisation [154]. Using treatment at different stages, CDF was shown to inhibit the initial sprouting and migration of LECs, processes dependent on Vegfc/Vegfr3 signalling. In agreement with this, they found that CDF reduced VEGF-C-dependent phospho-ERK levels in venous endothelial cells (sources of LECs (Figure 1B)) in zebrafish and in cultured adult human dermal microvascular lymphatic endothelial cells (HMVEC-dLy-Ad). CDF also inhibited excessive endothelial cell proliferation in a genetic *vegfc* overexpression model [7] and, in a separate study, was used to inhibit Vegfr-dependent compensatory lymphangiogenesis in zebrafish [125]. Unlike pan-VEGFR inhibitors such as sunitinib malate, CDF is not an inhibitor of VEGFR3 kinase activity, but is capable of prolonged inhibition of lymphangiogenesis following a brief 12 h treatment [154]. This suggests that CDF may act through a unique target to inhibit Vegfc/Vegfr3 signalling, warranting further investigation. Toluquinol, a natural compound isolated from a marine fungus, was also shown to inhibit TD development in zebrafish (Table 2) [155]. Toluquinol decreased viability, migration, tube formation, and sprouting of cultured HMVEC-dLy-Ad, and lymphatic outgrowth in cultured mouse thoracic duct explants. Further, toluquinol inhibited VEGF-C-induced infiltration of lymphatic vessels into sponge ear implants and in cornea following cauterization in mice. Mechanistically, toluquinol reduces VEGF-C-induced phosphorylation of VEGFR3 and its downstream effectors AKT and ERK1/2 in HMVEC-dLy-Ad. The consistency of results obtained across zebrafish and mammalian models in the presented studies show that anti-lymphangiogenic drugs discovered using zebrafish developmental lymphangiogenesis as a readout can be readily translated to mammalian applications.

## 7. Limitations of Using Zebrafish to Model Induced Lymphangiogenesis

Zebrafish models of induced lymphangiogenesis provide a uniquely accessible platform that complements mammalian models for studying mechanisms and therapeutics for conditions associated with induced lymphangiogenesis. However, their relative infancy as a model introduces limitations that need to be taken into consideration. First, the majority of models highlighted above use pre-adult zebrafish (less than 3 months old), which are still undergoing development (Table 1). While this may be beneficial for modelling childhood diseases (e.g., congenital lymphatic anomalies [30,31]), there may be differences in molecular and cellular mechanisms that drive induced lymphangiogenesis between developing and mature adult vessels. Additionally, zebrafish embryos, larvae, and juveniles under 3 weeks of age do not yet possess a fully functional adaptive immune system [156,157]. Consequently, models using zebrafish at these stages (e.g., inflammatory lymphangiogenesis [124], and lymphatic disease models [28,30,31,129]) are unsuitable for studying the pro-lymphangiogenic roles of lymphocytes, which have been shown to promote lymphangiogenesis in various conditions [158,159,160]. Zebrafish models at later developmental or adult stages present challenges for conducting medium- to high-throughput experiments, such as drug screening for identifying potential therapeutics for diseases associated with induced lymphangiogenesis [129,153,154]. While Castranova and colleagues have established methodologies for live imaging lymphatic vessels and their interaction with immune cells in adult fish [86,87], these approaches are not yet widely used due to significant challenges associated with live imaging adult fish (e.g., the requirement for intubation to provide flowing water to fish while imaging). These limitations of using adult zebrafish for research significantly reduces some of the unique advantages of using the zebrafish model. Nevertheless, studies should also develop induced lymphangiogenesis models using late-stage juveniles or adult zebrafish to more accurately model the pathophysiology of human diseases and conditions associated with induced lymphangiogenesis. Second, although zebrafish share a high degree of genetic and physiological conservation with humans [57], it is not a mammal, and therefore some difference may exist between mammalian and zebrafish lymphangiogenesis. For example, there are differences in Vegf-Vegfr binding capabilities: zebrafish Vegfr3 (Flt4) does not bind to both Vegfc and Vegfd, whereas human VEGFR3 does [161]. While this distinction has facilitated the identification of organotypic modes of lymphangiogenesis in zebrafish, it also underscores the importance of validating mechanisms and efficacies of therapeutics and therapeutic targets identified in zebrafish models using human endothelial cells or mammalian in vivo models. Finally, the extent to which zebrafish lymphatic vessels intersect with the secondary vascular system is not fully characterised. This is particularly relevant when using adult zebrafish as models for induced lymphangiogenesis, as secondary vessels are present and functional in adult zebrafish [145,146]. A more thorough characterisation of the interaction between lymphatic vessels and secondary vessels, and whether lymphatic vessels can readily transdifferentiate into secondary vessels during regeneration or pathology in late juvenile or adult zebrafish, is required.

## 8. Conclusions and Future Directions

Although the number of zebrafish induced lymphangiogenesis models is still low, the list of models has rapidly expanded in recent years and is expected to continue to grow. For example, models of cancer-associated lymphangiogenesis using later-stage zebrafish, when functional lymphatic vessels have developed, are currently lacking and represent a significant opportunity for development. The future expansion in the number of zebrafish models of induced lymphangiogenesis will be driven by advancements in zebrafish research tools and methodologies (e.g., new lymphatic-specific drivers, transgenics, disease models, and drug treatment/live imaging approaches [83,108,112,162,163,164,165,166]). Of note, lymphatic-specific enhancers are increasingly being used to generate transgenic lines that specifically label lymphatic vessels or valves (e.g., *Tg*(*FRT-Xla.Actc1:DsRed-GAB-FRT,LOXP-en.cdh6-LOXP-gata2a:EGFP-5HS4*)*^uom105^*) [82,112,147,167]. These transgenic lines allow clearer differentiation of lymphatic vessels and valves compared to traditionally used lymphatic reporter lines driven by promoters of genes such as *lyve1b*, *prox1a*, and *mrc1a* [81,85,111], as these also label other cell types such as veins and muscle. Therefore, these lymphatic-specific enhancer-driven lines could facilitate a more detailed characterisation of induced lymphangiogenesis in various contexts. However, the LEC/lymphatic valve specificity of these enhancer lines remains to be validated across different models of induced lymphangiogenesis. The various zebrafish models of induced lymphangiogenesis discussed in this review have led to novel discoveries in disease pathology, while also highlighting promising, clinically relevant therapeutic strategies. When combined with mammalian models of induced lymphangiogenesis, the zebrafish model is poised to play a pivotal role in the rapid and personalised identification and characterisation of novel anti- and pro-lymphangiogenic therapeutic strategies for various human diseases.

## Figures and Tables

**Figure 1 pharmaceuticals-18-01076-f001:**
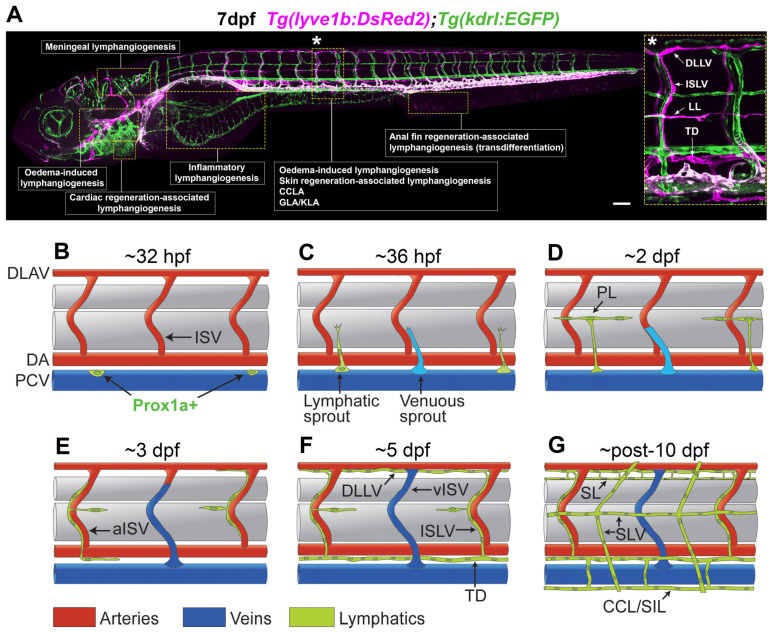
(**A**) Confocal lateral image of a 7 days post-fertilisation (dpf) *Tg*(*-5.2lyve1b:DsRed2*)*^nz101^*;*Tg*(*kdrl:EGFP*)*^s843^* zebrafish larvae, showing blood vessels in green and lymphatic vessels and veins in magenta. Regions where induced lymphangiogenesis can be observed in reviewed models are outlined. The asterisk (*) labels the position of inset showing magnified and labelled trunk lymphatic vessels, including the thoracic duct (TD), intersegmental lymphatic vessels (ISLV), lateral lymphatics (LL), and the dorsal longitudinal lymphatic vessel (DLLV). Note that for some models that use adult zebrafish, the outlines point to relative positions in the larvae. (**B**–**G**) Schematic representation showing developmental stages of trunk lymphatic vessels. At approximately 32 h post-fertilisation (hpf), a subset of venous endothelial cells in the posterior cardinal vein (PCV) begin to express Prox1a (**B**). These Prox1a-positive endothelial cells sprout from the PCV at approximately 36 hpf (**C**) and give rise to parachordal LECs (PLs) at the horizontal myoseptum by 2 dpf (**D**). Venous sprouting also happens at the same time (**C**), and these venous sprouts anastomose with intersegmental vessels (ISVs, (**D**)) to form venous ISVs (vISVs, (**E**,**F**)). PLs continue to proliferate and migrate along arteries such as the arterial ISVs (aISVs), dorsal aorta (DA), and the dorsal longitudinal anastomotic vessel (DLAV) (**E**) to eventually form the trunk lymphatic vessels (such as the TD, ISLV, and DLLV, (**F**)) by 5 dpf. By approximately 10 dpf, spinal lymphatics (SL), collateral cardinal lymphatics (CCL, also known as supraintestinal lymphatics (SIL)), and superficial lymphatic vessels (SLV) start to form (**G**). Scale bar indicates 100 μm.

**Table 1 pharmaceuticals-18-01076-t001:** A list of induced lymphangiogenesis models in zebrafish.

Model of Induced Lymphangiogenesis	Induction Method	Developmental Stage	Reference
Cancer-associated lymphangiogenesis	Mouse melanoma cell xenotransplantation	3 dpf	[123]
Inflammatory lymphangiogenesis	Treatment with enterocolitic agents DSS or TNBS	7 dpf	[124,130]
Meningeal lymphangiogenesis	Intracranial human VEGF-C recombinant protein injection	50 dpf	[86]
Compensatory lymphangiogenesis	Osmotic shock by 24 h incubation in hypertonic medium	4–9 dpf	[125]
Cardiac regeneration-associated lymphangiogenesis	Heart injury (resection, cryoinjury, cell ablation)	3–6 mpf	[119,121,126,127]
Anal fin regeneration-associated lymphangiogenesis	Anal fin resection	3–6 mpf	[108]
Skin regeneration-associated lymphangiogenesis	Removal of scales and scraping of skin with a dissecting knife	6–8 mpf	[87]
Central conductive lymphatic anomaly model	Injection of *ephb4*-targeting morpholino at single-cell stage	4 dpf	[128]
	Mosaic expression of *mrc1a*-driven KRAS p.G12D or p.G13D by construct injection at single-cell stage	5 dpf	[30]
	Mosaic expression of *mrc1a*-driven ARAF p.S214P by construct injection at single-cell stage	7 dpf	[31]
Generalised lymphatic anomaly/Kaposiform lymphangiomatosis model	4-hydroxytamoxifen-inducible *lyve1b*-driven expression of NRAS p.Q61R at 24 hpf	5 dpf	[129]

hpf—hours post-fertilisation; dpf—days post-fertilisation; mpf—months post-fertilisation.

## Data Availability

No new data were created or analyzed in this study. Data sharing is not applicable to this article.

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
