# Peer review of "Zebrafish Models of Induced Lymphangiogenesis: Current Advancements and Therapeutic Discovery"

_pharmaceuticals, 2025, doi:10.3390/ph18071076_

Round 1
Reviewer 1 Report
Comments and Suggestions for Authors
Srdjan Boskovic and Kazuhide Shaun Okuda submitted an interesting review about the zebra fish models of lymphangiogenesis. The topic was of a certain significance, but less frequently discussed in this field. The manuscript could be considered for publication after a Minor Revision. Detailed comments:
- According to the length of the review, it might better stand as a Mini-Review. Please indicate it in the Title.
- Some brief introduction to zebra fish models for other diseases could be added.
- A scheme or photograph to show the grow and development stages of the lymphatic system of zebra fish could be added.
- Zebra fish model was suitable for bioimaging study after nanoparticle administration. The authors were recommended to add relevant discussion.
- Please denote a separate Conclusion Section at the end of the manuscript.
Author Response
We sincerely thank all reviewers for their thoughtful and constructive comments, which have greatly contributed to improving the quality of our manuscript. We have carefully addressed each point raised and made the corresponding revisions. We believe that these changes have significantly strengthened the clarity and impact of our review. We have highlighted corrected/modified parts in red in the resubmitted review. Specific responses to each comments as below.
Comment 1: According to the length of the review, it might better stand as a Mini-Review. Please indicate it in the Title.
Response: To address all reviewer’s comments, we have significantly expanded the content of our review, which we believe now meets the scope and length of a full review article.
Comment 2: Some brief introduction to zebra fish models for other diseases could be added.
Response: Thank you for this great suggestion. We have added a sentence in the Introduction (see Lines 101–103) listing zebrafish models of diseases, including cardiovascular, cancer, neurodegenerative, infectious, and muscular disorders."
Comment 3: A scheme or photograph to show the grow and development stages of the lymphatic system of zebra fish could be added.
Response: This is a great suggestion, and we have now added Figures 1B–G, which illustrates the development of trunk lymphatic vessels in zebrafish, with corresponding figure legends explaining each step. Additionally, we have included a section in the main text describing the development of zebrafish lymphatic vessels (see Lines 125–142).
Comment 4: Zebra fish model was suitable for bioimaging study after nanoparticle administration. The authors were recommended to add relevant discussion.
Response: Thank you for this comment. We agree that zebrafish is an emerging model for nanoparticle administration and research. However, as this technique has not yet been applied in the context of induced lymphangiogenesis in zebrafish, we considered it beyond the scope of this review.
Comment 5: Please denote a separate Conclusion Section at the end of the manuscript.
Response: We agree with this suggestion, and the review now includes a dedicated Conclusions and Future Directions section (see Lines 436-461).
We have also updated Figure 1A using a transgenic line that show zebrafish lymphatic and blood vessels more clearly (Tg(kdrl:EGFP);Tg(lyve1b:DsRed2)).
Reviewer 2 Report
Comments and Suggestions for Authors
In this study, the authors aimed to explain how zebrafishmodels are used to study induced lymphangiogenesis—theformation of new lymphatic vessels—in various physiologicaland pathological contexts. Due to their genetic similarity tohumans, optical transparency, and suitability for imaging andgenetic manipulation, zebrafish serve as powerful tools formodeling lymphatic diseases and identifying therapeutictargets. The article explores mechanisms of lymphangiogenesis in conditions such as cancer, inflammation, edema, and tissue regeneration, and highlightsdrug discovery efforts using zebrafish. It also addresses thelimitations of this model and emphasizes its translationalrelevance to human disease research and potential clinicalapplications. The manuscript is well designed and presented in an acceptable format; but there are some issues that need to be resolved before it can be published:
• The manuscript is overall well-written with appropriateacademic tone.
• Terminology is consistent and discipline-specific, whichsupports technical accuracy.
• There are minor grammatical and typographical issues, mainly due to PDF formatting
• Some sentences, especially in the Introduction andDiscussion, are overly long and complex, affecting readability.
• The confocal image is valuable and informative.
• The manuscript provides a thorough and technically solidreview of zebrafish models of induced lymphangiogenesis.
• The writing assumes a high level of prior knowledge in vascular biology and zebrafish genetics.
• Some points (e.g., limitations of using early-stage zebrafishlacking adaptive immunity) are repeated unnecessarily.
• The text could benefit from more concise structuring toavoid redundancy.
• The conclusion is embedded within the Discussion section.
• While the content is appropriate, separating the conclusioninto a distinct final paragraph or section would improveemphasis and reader take-away.
Author Response
We sincerely thank all reviewers for their thoughtful and constructive comments, which have greatly contributed to improving the quality of our manuscript. We have carefully addressed each point raised and made the corresponding revisions. We believe that these changes have significantly strengthened the clarity and impact of our review. We have highlighted corrected/modified parts in red in the resubmitted review. Specific responses to each comments as below.
Comments 1:
There are minor grammatical and typographical issues, mainly due to PDF formatting.
Some sentences, especially in the Introduction and Discussion, are overly long and complex, affecting readability.
The text could benefit from more concise structuring to avoid redundancy.
Some points (e.g., limitations of using early-stage zebrafishlacking adaptive immunity) are repeated unnecessarily.
Response: We apologise for these oversights. We have thoroughly reviewed the manuscript and corrected them to the best of our ability.
Comment 2: The confocal image is valuable and informative.
Response: Thank you for this comment. We have now updated Figure 1A using a transgenic line that show zebrafish lymphatic and blood vessels more clearly (Tg(kdrl:EGFP);Tg(lyve1b:DsRed2)).
Comment 3: The writing assumes a high level of prior knowledge in vascular biology and zebrafish genetics.
Response: We agree that more background information on how zebrafish lymphatic vessels develop would significantly help readers new to the field. We have now included a detailed section describing this process (see Lines 125–142), along with schematic images (and corresponding figure legends) to aid understanding (Figure 1B-G).
Comments 4:
The conclusion is embedded within the Discussion section.
While the content is appropriate, separating the conclusion into a distinct final paragraph or section would improve emphasis and reader take-away.
Response: We agree with this suggestion, and the review now includes a dedicated Conclusions and Future Directions section.
Reviewer 3 Report
Comments and Suggestions for Authors
The manuscript "Zebrafish Models of Induced Lymphangiogenesis: Current Advancements and Therapeutic Discovery" aimed to gather and highlight studies for which zebrafish was a test model to study induced lymphangiogenesis, as well as discuss how this model is being used to identify potentially clinically relevant therapeutic leads. The authors gathered 14 different studies for which zebrafish was used as test model for induced lymphangiogenesis and discussed these works in detail. Moreover, these works were divided in different subsections according to different processes/endpoints in zebrafish. Also, one section details some data about anti-lymphangiogenic drug discovery for which zebrafish was the test model. Also, the authors discussed the limitations and future challenges for the use of zebrafish as a model in this field.
Overall, the manuscript is well-written and organized. I aknowledge the detail in the report and discussion of the cited literature, regarding the use of zebrafish as a test model for induced lymphangiogenesis. The introduction is also clear, pertinent and for the most part really detailed. I also recognize the attention to the limitations that were clearly indicated and discussed. Nonetheless, I believe that the manuscript needs some work to be publishable. My main concerns/suggestions are the following:
1) The manuscript lacks a section for the applied methodology for this research. The authors should indicate how the research was conducted (e.g. databases, keywords, etc). Please include a methodology section in the manuscript
2) In Lines 100-116, zebrafish lymphatic system is briefly described. I believe, this part could be more detailed. For example, the development of the larval lymphatic system until the adult animal should be described, since the existing literature is focused on pre-adult stages of development (as stated in the limitations). The whole description of zebrafish lymphatic system and lymphangiogenesis, could have a dedicated section for itself.
3) In table 1, I suggest to include the age of the animals in the developmental stage collumn. The term "larvae" or "juvenile" is too vague.
4) Also for the tables, I suggest the authors to reference the table more often in the text. I was a bit lost, trying to link the text with the corresponding table.
5) The section about "Anti-lymphangiogenic drug discovery using zebrafish" should contain more detail. Comparing to other sections, the text is shorter and less detailed. Moreover, the identification of potentially clinically relevant therapeutic leads is one of the aims of this review, so this section should be more developed.
6) In the limitations, I believe that should be clearly stated that the number of works using zebrafish as test model is low. After reading the previous sections, this was the first conclusion that I had and this was not clearly stated in the limitations section. Also, in this section I suggest the authors to include a brief mention about the most suitable zebrafish lineages (e.g transgenic Tg(fli1a:egfp)y1 line) to be used as test models.
7) I suggest the authors to write the references in a crescent numerical order. E.g. Line 118: [91–93,86,94–97,30,31,98,99,66,65], is confusing. Please, correct this.
8) There is a typo in the reference 53. The number 53 is repeated. Please correct it.
Author Response
We sincerely thank all reviewers for their thoughtful and constructive comments, which have greatly contributed to improving the quality of our manuscript. We have carefully addressed each point raised and made the corresponding revisions. We believe that these changes have significantly strengthened the clarity and impact of our review. We have highlighted corrected/modified parts in red in the submitted review. Specific responses to each comments as below.
Comment 1: The manuscript lacks a section for the applied methodology for this research. The authors should indicate how the research was conducted (e.g. databases, keywords, etc). Please include a methodology section in the manuscript
Response: Thank you for this comment. We have not conducted a specific search of literature using keywords. However, we have added a “Survey of literature:” section to clarify how we conducted our literature search for the manuscript (see Lines 463-466).
Comment 2: In Lines 100-116, zebrafish lymphatic system is briefly described. I believe, this part could be more detailed. For example, the development of the larval lymphatic system until the adult animal should be described, since the existing literature is focused on pre-adult stages of development (as stated in the limitations). The whole description of zebrafish lymphatic system and lymphangiogenesis, could have a dedicated section for itself.
Response: We sincerely thank the reviewer for this suggestion, which has significantly improved our review. We have included a section in the main text that briefly describes the development of zebrafish lymphatic vessels through to adulthood (see Lines 125–142), and we have added Figures 1B–G, which illustrate the development of trunk lymphatic vessels in zebrafish, with corresponding figure legends explaining each step.
Comment 3: In table 1, I suggest to include the age of the animals in the developmental stage collumn. The term "larvae" or "juvenile" is too vague.
Response: We agree with the reviewer’s comment and apologise for the vagueness. We have now added the exact ages of zebrafish used for each induced lymphangiogenesis model (see updated Table 1).
Comment 4: Also for the tables, I suggest the authors to reference the table more often in the text. I was a bit lost, trying to link the text with the corresponding table.
Response: Thank you for this comment, and we apologise for any confusion caused. We have now added more references to the Table in the main text of the review.
Comment 5: The section about "Anti-lymphangiogenic drug discovery using zebrafish" should contain more detail. Comparing to other sections, the text is shorter and less detailed. Moreover, the identification of potentially clinically relevant therapeutic leads is one of the aims of this review, so this section should be more developed.
Response: We agree with the reviewer’s comments and have significantly expanded this section (see Lines 338-391).
Comment 6: In the limitations, I believe that should be clearly stated that the number of works using zebrafish as test model is low. After reading the previous sections, this was the first conclusion that I had and this was not clearly stated in the limitations section. Also, in this section I suggest the authors to include a brief mention about the most suitable zebrafish lineages (e.g transgenic Tg(fli1a:egfp)y1 line) to be used as test models.
Response: We thank the reviewer for this important comment. We agree that the number of zebrafish induced lymphangiogenesis models is currently limited. However, we view this as more of an opportunity than a limitation (possibility of generating additional zebrafish induced lymphangiogenesis models) and have therefore included this perspective in the Conclusions and Future Directions section (see Lines 438–442). Additionally, we have added a sentence describing recently developed lymphatic-specific enhancer lines that specifically label lymphatic vessels and valves. We believe these lines should be used more extensively in future research in this field, rather than relying solely on commonly used lines such as the lyve1b:DsRed2 transgenic. This is now added in the Conclusions and Future Directions section as well (see Lines 445–455).
Comments 7:
I suggest the authors to write the references in a crescent numerical order. E.g. Line 118: [91–93,86,94–97,30,31,98,99,66,65], is confusing. Please, correct this.
There is a typo in the reference 53. The number 53 is repeated. Please correct it.
Response: We apologise for these oversights, which have now been corrected.
We have also updated Figure 1A using a transgenic line that show zebrafish lymphatic and blood vessels more clearly (Tg(kdrl:EGFP);Tg(lyve1b:DsRed2)).
Round 2
Reviewer 2 Report
Comments and Suggestions for Authors
The manuscript has been improved significantly and can now be accepted for publication
Reviewer 3 Report
Comments and Suggestions for Authors
The revised version of the manuscript "Zebrafish Models of Induced Lymphangiogenesis: Current Advancements and Therapeutic Discovery" was significantly improved. I aknowledge the effort of the authors to accomodate the critics and suggestions of the reviewers.